# Deep Sparse Latent Feature Models for Knowledge Graph Completion

## Abstract

Recent progress in knowledge graph completion (KGC) has focused on text-based approaches to address the challenges of large-scale knowledge graphs (KGs). Despite their achievements, these methods often overlook the intricate interconnections between entities, a key aspect of the underlying topological structure of a KG. Stochastic blockmodels (SBMs), particularly the latent feature relational model (LFRM), offer robust probabilistic frameworks that can dynamically capture latent community structures and enhance link prediction. In this paper, we introduce a novel framework of sparse latent feature models for KGC, optimized through a deep variational autoencoder (VAE). Our approach not only effectively completes missing triples but also provides clear interpretability of the latent structures, leveraging textual information. Comprehensive experiments on the WN18RR, FB15k-237, and Wikidata5M datasets show that our method significantly improves performance by revealing latent communities and producing interpretable representations.

## 1 Introduction

The majority of real-world phenomena exhibit multifaceted characteristics. For instance, social networks are not merely a collection of isolated individuals but represent a complex web of interactions across various contexts. Knowledge graphs (KGs) organize information into triples $(h, r, t)$, where $h$ denotes the head entity, $t$ the tail entity, and $r$ the relationship, forming extensive semantic networks. However, real-world KGs like DBpedia (Auer et al., 2007) and Wikidata (Vrandečić & Krötzsch, 2014) often suffer from incompleteness, missing key entities and relationships (Dong et al., 2014). Knowledge graph completion (KGC) aims to infer this missing information, improving the utility and completeness of the graph.

Early developments in KGC centered around knowledge graph embedding (KGE) techniques (Bordes et al., 2013; Sun et al., 2019; Balažević et al., 2019), which focused on learning low-dimensional embeddings for entities and relations, applying various scoring functions to triples. More recently, text-based methods (Yao et al., 2019; Wang et al., 2021a; 2022) utilizing pre-trained language models (PLMs) have achieved state-of-the-art performance on large-scale datasets such as Wikidata5M (Wang et al., 2021b). These approaches generally rely on the transformation of the head embedding **h** into the tail **t** through the relation **r**, **yet the complex interconnectivity among communities associated with entities remains insufficiently exploited**. As highlighted by Stanley et al. (2019), network topologies typically exhibit dense connections within groups and fewer connections between them. Inspired by this, we approach triple completion from a broader perspective by focusing on relational connections across entity communities. For instance, in the KG depicted in Figure 1, entities are categorized into overlapping communities. To answer the query regarding the relationship between Michael Jordan and Gregg Popovich, 'acquaintance' emerges as a plausible candidate, which can be inferred from the interconnections observed between the two communities—'NBA Players' and 'NBA Coaches'.

Uncovering the latent structure of graph data is a key area of focus in statistical network analysis (Porter et al., 2009; Latouche et al., 2010). Stochastic blockmodels (SBMs) (Airoldi et al., 2008; Miller et al., 2009; Latouche et al., 2011) are a widely recognized class of probabilistic models that assign cluster memberships to graph nodes, and are highly regarded in both academic and industrial settings. A notable variant is the latent feature relational model (LFRM), a type of overlapping stochastic blockmodel (OSBM), which allows nodes to belong to multiple groups and leverages

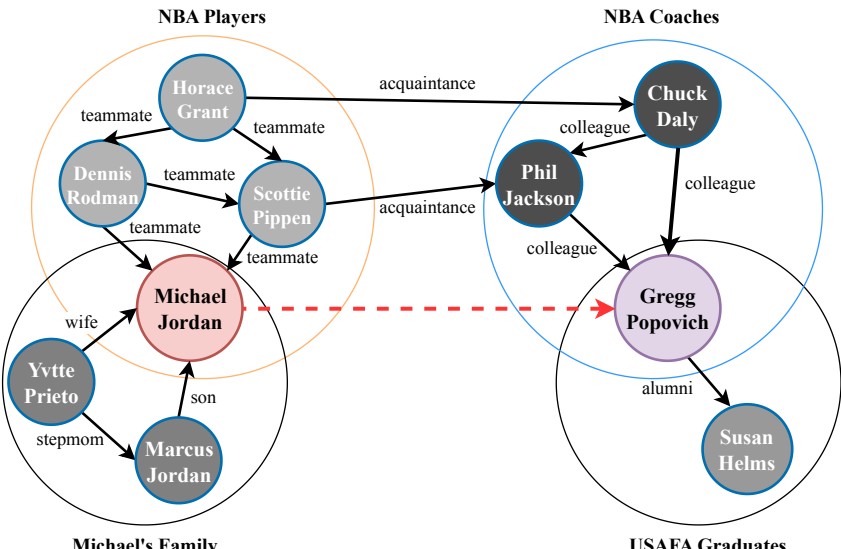

Figure 1: A simplified example of KG involving diverse communities. Solid black arrows indicate existing links, while the dashed red arrow represents a missing link for a KGC model to predict. Each community within the graph is encircled, highlighting the overlapped groups of interconnected entities.

an Indian Buffet Process (IBP) prior on the node-community assignment matrix $Z$ to discover the number of latent communities. These models typically rely on MCMC (Miller et al., 2009) or variational inference (Zhu et al., 2016) to infer latent variables. While DGLFRM (Mehta et al., 2019) enhances SBM inference using a deep sparse variational autoencoder (VAE) (Kingma & Welling, 2013), **it is not tailored for KGC tasks and faces challenges when scaled to large graphs with hundreds of thousands or even millions of nodes**.

**Contributions**. We propose DSLFM-KGC, a novel method for tackling the KGC challenge by utilizing latent community structures in KGs. Our main contributions are as follows: **i)** we design an end-to-end probabilistic model for KGC that integrates additional sparse clustering features into triple representation, implemented through a deep VAE (Kingma & Welling, 2013); **ii)** DSLFM-KGC provides robust performance and interpretability in completing missing triples by leveraging community-level interconnections in entities; and **iii)** the deep architecture allows for scalable inference. Through extensive experiments on the WN18RR, FB15k-237, and Wikidata5M datasets, **iv)** we showcase our model's superior capability and scalability in managing KGC tasks and uncovering interpretable latent structures.

## 2 PRELIMINARIES

### 2.1 LATENT FEATURE RELATIONAL MODEL

The SBMs (Holland et al., 1983; Airoldi et al., 2008; Miller et al., 2009) are fundamental approaches for analyzing relational data, where a graph with $N$ nodes is represented by a binary adjacency matrix $A \in \{0,1\}^{N \times N}$. In this matrix, $A_{i,j} = 1$ indicates a link between node $i$ and node $j$. Each node $i$ in an SBM is associated with a one-hot latent variable $\mathbf{z}_i \in \{0,1\}^K$ to indicate its community membership, where $K$ is the number of node communities.

For scenarios where nodes belong to multiple communities, the OSBM (Latouche et al., 2011) adapts the latent indicator $\mathbf{z}_i$ into a multivariate Bernoulli vector consisting of $K$ independent Bernoulli variables, denoted as $\mathbf{z}_i \sim \mathcal{MB}(\mathbf{z}|\boldsymbol{\pi})$:

$$\mathcal{MB}(\mathbf{z}|\boldsymbol{\pi}) = \prod_{k=1}^{K} \text{Bernoulli}(z_k|\pi_k) = \prod_{k=1}^{K} \pi_k^{z_k}(1-\pi_k)^{1-z_k} \tag{1}$$

where $\pi_k \in [0, 1]$. The link probability between two nodes in OSBM is defined as a bilinear function of their indicator vectors:

$$p(A_{i,j} = 1|\mathbf{z}_i, \mathbf{z}_j, W) = \sigma(\mathbf{z}_i^\top W \mathbf{z}_j) \tag{2}$$

Here, $W$ is a real-valued $K \times K$ matrix, with $w_{kl}$ influencing the link likelihood between communities $k$ and $l$, and $\sigma(\cdot)$ is the sigmoid function.

Expanding on OSBM, the LFRM integrates the IBP prior (Miller et al., 2009) on the binary node-community matrix $Z = [\mathbf{z}_1, \ldots, \mathbf{z}_N]^\top$, enabling dynamic learning of the number of communities. Traditional inference methods used in SBMs, such as MCMC or variational inference, often struggle to scale in large networks. To overcome this, DGLFRM (Mehta et al., 2019) uses a VAE (Kingma & Welling, 2013), employing a graph convolutional network (GCN) (Kipf & Welling, 2016) to encode the variational distribution $q(Z)$ and a non-linear multilayer perceptron (MLP) to model the link probability $p(A_{i,j}|\mathbf{z}_i, \mathbf{z}_j, W)$. Despite its advances, DGLFRM encounters difficulties when applied to large-scale heterogeneous KGs, which feature entities and relations of diverse types.

## 2.2 KNOWLEDGE GRAPH COMPLETION

A KG is commonly defined as $\mathcal{G} = (\mathcal{E}, \mathcal{R}, \mathcal{T})$, where $\mathcal{E}$ is the set of entities, and $\mathcal{R}$ is the set of relations. The set $\mathcal{T} = \{(h, r, t)|h, t \in \mathcal{E}, r \in \mathcal{R}\}$ contains factual triples, each representing a directed labeled edge $h \xrightarrow{r} t$ in the KG. Furthermore, modern KGs often include meta-information $\mathcal{M}$, such as natural language descriptions (Yao et al., 2019; Wang et al., 2022) or multi-modal data (Zhang et al., 2024a). For any entity $e \in \mathcal{E}$ and any relation $r \in \mathcal{R}$, $\mathcal{M}(e)$ and $\mathcal{M}(r)$ denote the corresponding meta-information.

For a given query $(h, r, ?)$, the task of KGC involves identifying the missing tail entity by retrieving the most plausible candidate $\hat{t}$ from the entity set $\mathcal{E}$, such that $(h, r, \hat{t})$ is valid. From the KGC perspective, we model a KG as containing a query set $\mathcal{Q} = \{(h, r)|h \in \mathcal{E}, r \in \mathcal{R}\}$, a candidate answer (entity) set $\mathcal{E}$, and a mapping $\mathcal{A} : \mathcal{Q} \times \mathcal{E} \rightarrow \{0, 1\}$ that identifies whether a query has a valid answer in the KG $\mathcal{G}$. This mapping is represented as a binary matrix $A \in \{0, 1\}^{|\mathcal{Q}| \times |\mathcal{E}|}$, analogous to an adjacency matrix, where $A_{hr,t} = 1$ if the triple $(h, r, t) \in \mathcal{T}$, and $A_{hr,t} = 0$ otherwise.

# 3 METHODOLOGY

This section presents the framework of DSLFM-KGC. We begin by describing the probabilistic generative model for KGs, emphasizing its application to KGC. Following this, we elaborate the VAE architecture employed for inference, detailing the design and implementation of both the encoder and decoder.

## 3.1 GENERATIVE MODEL

We assume that triples within a KG are conditionally independent, given their latent communities. The generation process of a KG unfolds as follows:

For each query $(h, r) \in \mathcal{Q}$ and each answer $t \in \mathcal{E}$, draw the membership indicator vectors:

$$\mathbf{z}_{hr} \sim \mathcal{MB}(\mathbf{z}|\boldsymbol{\pi}_{hr}), \ \mathbf{z}_t \sim \mathcal{MB}(\mathbf{z}|\boldsymbol{\pi}_t) \tag{3}$$

Next, draw the latent feature vectors:

$$\mathbf{w}_{hr} \sim \mathcal{N}(\mathbf{w}|\mathbf{0}, \sigma^2\mathbf{I}), \ \mathbf{w}_t \sim \mathcal{N}(\mathbf{w}|\mathbf{0}, \sigma^2\mathbf{I}) \tag{4}$$

finally, draw the triple:

$$A_{hr,t} \sim p(A_{hr,t}|\mathbf{z}_{hr}, \mathbf{z}_t, \mathbf{w}_{hr}, \mathbf{w}_t) \tag{5}$$

Here, $\mathbf{z}_{hr} \in \{0, 1\}^{K_1}, \mathbf{z}_t \in \{0, 1\}^{K_2}$ are binary vectors with elements equal to one indicating their respective community memberships, and $\mathbf{w}_{hr} \in \mathbb{R}^{K_1}, \mathbf{w}_t \in \mathbb{R}^{K_2}$ represent the strength of their community memberships, *i.e.*, latent features. Typically, query clusters outnumber entity clusters due to the diversity of entity-relation pairs.

The distribution $p(A_{hr,t}|\mathbf{z}_{hr}, \mathbf{z}_t, \mathbf{w}_{hr}, \mathbf{w}_t)$ is modeled as a Bernoulli distribution, with the probability $p(A_{hr,t} = 1|\mathbf{z}_{hr}, \mathbf{z}_t, \mathbf{w}_{hr}, \mathbf{w}_t)$ signifying that the answer aligns with the query, thus confirming the existence of the triple $(h, r, t)$ in the KG:

$$\mathbf{f}_{hr} = \mathbf{w}_{hr} \odot \mathbf{z}_{hr}, \ \mathbf{f}_t = \mathbf{w}_t \odot \mathbf{z}_t \tag{6}$$

$$p(A_{hr,t} = 1|\mathbf{z}_{hr}, \mathbf{z}_t, \mathbf{w}_{hr}, \mathbf{w}_t) = \sigma\left(\mathbf{f}_{hr}^\top \mathbf{f}_t\right) \tag{7}$$

where $\odot$ is the Hadamard product.

Let $Z_{\text{qry}}$ and $Z_{\text{ans}}$ denote the membership indicator matrices for queries and answers, respectively, and let $W_{\text{qry}}$ and $W_{\text{ans}}$ denote the membership strength matrices. Then, $F_{\text{qry}} = Z_{\text{qry}} \odot W_{\text{qry}}$ and $F_{\text{ans}} = Z_{\text{ans}} \odot W_{\text{ans}}$ constitute a sparse latent feature model (Ghahramani & Griffiths, 2005; d'Aspremont et al., 2004; Jolliffe et al., 2003). We use the Indian Buffet Process (IBP) (Griffiths & Ghahramani, 2011) prior on the indicator matrices to facilitate the learning of the number of communities, thereby establishing an infinite latent feature model (Ghahramani & Griffiths, 2005).

$$Z_{\text{qry}} \sim \mathcal{IBP}(\alpha_{\text{qry}}), \ Z_{\text{ans}} \sim \mathcal{IBP}(\alpha_{\text{ans}}) \tag{8}$$

## 3.2 VAE ENCODER

We adopt the stick-breaking construction of the IBP (Teh et al., 2007) to model $\mathbf{z}_{hr}$:

$$v_{hr,k} \sim \text{Beta}(\alpha_{\text{qry}}, 1), \ k = 1, \ldots, K_1$$

$$\pi_{hr,k} = \prod_{j=1}^{k} v_{hr,j}, \ z_{hr,k} \sim \text{Bernoulli}(\pi_{hr,k}) \tag{9}$$

The sampling of $\mathbf{z}_t$ can be achieved similarly. By employing the stick-breaking approach, the effective number of communities engaged can be learned by setting a sufficiently large truncation level $K = K_1 = K_2$ in our model.

Let $\mathcal{H} = \{V_{\text{qry}}, Z_{\text{qry}}, W_{\text{qry}}, V_{\text{ans}}, Z_{\text{ans}}, W_{\text{ans}}\}$ denote the set of latent variables and $\mathcal{O} = \{\mathcal{Q}, \mathcal{E}, A\}$ the set of observations. We utilize an encoder network to approximate the true posterior $p(\mathcal{H}|\mathcal{O})$ with a variational distribution $q_\phi(\mathcal{H})$ parameterized by $\phi$, which is factorized following the mean-field approximation:

$$q_\phi(\mathcal{H}) = \prod_{k=1}^{K} \left( \prod_{(h,r)\in\mathcal{Q}} q_\phi(v_{hr,k})q_\phi(z_{hr,k})q_\phi(w_{hr,k}) \prod_{t\in\mathcal{A}} q_\phi(v_{t,k})q_\phi(z_{t,k})q_\phi(w_{t,k}) \right) \tag{10}$$

The distributions involved are defined as follows:

$$q_\phi(v_{hr,k}) \triangleq \text{Beta}(c_{hr,k}, d_{hr,k}), \ k = 1, \ldots, K \tag{11}$$

$$q_\phi(z_{hr,k}) \triangleq q_\phi(z_{hr,k}|\mathcal{Q}) = \text{Bernoulli}(\pi_{hr,k}(\mathcal{Q})), \ k = 1, \ldots, K \tag{12}$$

$$q_\phi(w_{hr,k}) \triangleq q_\phi(w_{hr,k}|\mathcal{Q}) = \mathcal{N}(\mu_{hr,k}(\mathcal{Q}), \sigma_{hr,k}^2(\mathcal{Q})), \ k = 1, \ldots, K \tag{13}$$

where $\pi_{hr,k}, \mu_{hr,k}$ and $\sigma_{hr,k}^2$ are outputs of the encoder network $h_\phi$, *i.e.*, $\{\boldsymbol{\pi}_{hr}, \boldsymbol{\mu}_{hr}, \boldsymbol{\sigma}_{hr}^2\}_{(h,r)\in\mathcal{Q}} = h_\phi(\mathcal{Q})$ with $\mathcal{Q}$ as the input. In experiment, we found that treating $\mathbf{c}_{hr}$ and $d_{hr}$ as part of the encoder parameters (instead of encoding them from the posterior) helps mitigate over-parameterization. We define $q_\phi(v_{t,k}), q_\phi(z_{t,k})$ and $q_\phi(w_{t,k})$ in a similar vein, with $\mathcal{Q}$ replaced by $\mathcal{E}$.

Following recent progress in text-based approaches for the KGC task (Yao et al., 2019; Wang et al., 2022), we employ the strategy that individually encodes the textual descriptions of queries and answers using two BERT (Devlin et al., 2019) encoders, sharing pre-trained weights, and applying mean pooling:

$$\mathbf{e}_{hr} = \text{Pool}(\text{BERT}_{\text{qry}}(\mathbf{x}_{hr})), \quad \mathbf{e}_t = \text{Pool}(\text{BERT}_{\text{ans}}(\mathbf{x}_t)) \tag{14}$$

Here, $\mathbf{x}_{hr}$ and $\mathbf{x}_t$ represent the textual descriptions of the query and the answer after tokenization, respectively. Subsequently, a multi-layer perceptron (MLP) is leveraged to project the textual encodings into the latent space:

$$\{\pi_{hr,k}, \mu_{hr,k}, \sigma_{hr,k}\}_{k=1}^{K} = \text{MLP}(\mathbf{e}_{hr}), \quad \{\pi_{t,k}, \mu_{t,k}, \sigma_{t,k}\}_{k=1}^{K} = \text{MLP}(\mathbf{e}_t) \tag{15}$$

Figure 2: An overview of our DSLFM-KGC framework during inference. Initially, the encoder network $h_\phi$ encodes the textual information of a triple ($\mathbf{x}_{hr}$ and $\mathbf{x}_t$) into posterior distributions, as defined in Equations 14 and 15. Latent variables (e.g., $\mathbf{z}_{hr}$ and $\mathbf{w}_{hr}$) are then sampled using reparameterization tricks (see Appendix A.4), after which the decoder $g_\theta$ generates representations for the query and answer ($\mathbf{g}_{hr}$ and $\mathbf{g}_t$).

It is important to note that our approach leverages latent structures encoded within the textual semantic space, which has demonstrated enhanced expressiveness for KGC tasks. Additionally, the flexibility of our model allows for the use of various types of encoders, such as a multi-modal one, to further enhance expressiveness. We plan to explore these possibilities in future research.

We denote the overall encoder network with parameters $\phi$ as $h_\phi$. Integrating textual inputs not only enhances the performance of our model, but also provides deeper insights into the latent structure. This allows for the exploration of the mined communities through the descriptions of the entities within, the benefits of which will be demonstrated in the experiment section.

### 3.3 VAE DECODER

We model the probability distribution $p_\theta$ through a decoder network $g_\theta$, parameterized by $\theta$. Given the latent variables $\mathbf{z}_{hr}, \mathbf{z}_t, \mathbf{w}_{hr}$ and $\mathbf{w}_t$, the decoder network generates a link $A_{hr,t} \sim p_\theta(A_{hr,t}|\mathbf{z}_{hr}, \mathbf{z}_t, \mathbf{w}_{hr}$ and $\mathbf{w}_t)$. We first computes the Hadamard product to obtain $\mathbf{f}_{hr}$ and $\mathbf{f}_t$, as outlined in Equation 6. An MLP with non-linear activations is subsequently employed to transform $\mathbf{f}_{hr}, \mathbf{f}_t$ into $\mathbf{g}_{hr}, \mathbf{g}_t$, respectively.

$$\mathbf{g}_{hr} = \text{MLP}(\mathbf{f}_{hr}), \quad \mathbf{g}_t = \text{MLP}(\mathbf{f}_t) \tag{16}$$

The inner product of these transformed vectors is then computed to represent the confidence level that the triple $(h, r, t)$ exists in the KG. The use of an MLP, as opposed to relying solely on a single Hadamard product, enables more expressive representations and improves overall performance.

The architecture of our model is depicted in Figure 2.

### 3.4 INFERENCE

We jointly update the encoder $h_\phi$ and the decoder $g_\theta$ by minimizing the negative of the evidence lower bound (ELBO):

$$\mathcal{L} = \sum_{(h,r)\in\mathcal{Q}} \{D_{\text{KL}}\left[q_\phi(\mathbf{v}_{hr})||p_\theta(\mathbf{v}_{hr})\right] + D_{\text{KL}}\left[q_\phi(\mathbf{z}_{hr})||p_\theta(\mathbf{z}_{hr}|\mathbf{v}_{hr})\right] + D_{\text{KL}}\left[q_\phi(\mathbf{w}_{hr})||p_\theta(\mathbf{w}_{hr})\right]\}$$

$$+ \sum_{t\in\mathcal{E}} \{D_{\text{KL}}\left[q_\phi(\mathbf{v}_t)||p_\theta(\mathbf{v}_t)\right] + D_{\text{KL}}\left[q_\phi(\mathbf{z}_t)||p_\theta(\mathbf{z}_t|\mathbf{v}_t)\right] + D_{\text{KL}}\left[q_\phi(\mathbf{w}_t)||p_\theta(\mathbf{w}_t)\right]\}$$

$$- \sum_{(h,r)\in\mathcal{Q}} \mathbb{E}_q\left[\log p_\theta(\mathbf{x}_{hr}|\mathbf{z}_{hr}, \mathbf{w}_{hr})\right] - \sum_{t\in\mathcal{E}} \mathbb{E}_q\left[\log p_\theta(\mathbf{x}_t|\mathbf{z}_t, \mathbf{w}_t)\right]$$

$$- \sum_{(h,r)\in\mathcal{Q}} \sum_{t\in\mathcal{E}} \mathbb{E}_q\left[\log p_\theta(A_{hr,t}|\mathbf{z}_{hr}, \mathbf{z}_t, \mathbf{w}_{hr}, \mathbf{w}_t)\right] \tag{17}$$

where $D_{\text{KL}}[q(\cdot)||p(\cdot)]$ is the KL divergence of the distributions $q(\cdot)$ and $p(\cdot)$.

To further enhance KGC performance, we express the triple completion term $\log p_\theta(A_{hr,t}|\mathbf{z}_{hr}, \mathbf{z}_t, \mathbf{w}_{hr}, \mathbf{w}_t)$ as a contrastive loss. The contrastive framework is renowned

for its capacity to learn expressive representations, as it aims to maximize the mutual information between the inputs and the outputs (Ben-Shaul et al., 2023; Hjelm et al., 2018; Gutmann & Hyvärinen, 2012). Specifically, we utilize the supervised contrastive loss (Li et al., 2023a; Khosla et al., 2020):

$$\log p_\theta(A_{hr,t}|\mathbf{z}_{hr}, \mathbf{z}_t, \mathbf{w}_{hr}, \mathbf{w}_t) = \frac{1}{|\mathcal{N}^+|} \sum_{t \in \mathcal{N}^+} \log \frac{e^{(S(\mathbf{g}_{hr}, \mathbf{g}_t)-\gamma)/\tau}}{e^{(S(\mathbf{g}_{hr}, \mathbf{g}_t)-\gamma)/\tau} + \sum_{t' \in \mathcal{N}^-} e^{(S(\mathbf{g}_{hr}, \mathbf{g}_{t'})-\gamma)/\tau}} \tag{18}$$

where $\mathcal{N}^+$ represents the set of positive entities of the query $(h, r, ?)$, and $\mathcal{N}^-$ denotes the set of negative samples, encompassing all other entities within the same batch (Chen et al., 2020). The variable $\gamma$ denotes the additive margin, $\tau$ the temperature and $S(\mathbf{g}_{hr}, \mathbf{g}_t) = \cos(\mathbf{g}_{hr}, \mathbf{g}_t) = \mathbf{g}_{hr}^\top \mathbf{g}_t/(||\mathbf{g}_{hr}|| \cdot ||\mathbf{g}_t||) \in [-1, 1]$ the cosine similarity score function.

We then optimize the objective using Stochastic Gradient Variational Bayes (SGVB) and mini-batch gradient descent (Kingma & Welling, 2013). Given a batch of triples $B \subset \mathcal{G}$, and let the decoded representations $\mathbf{g}_{hr}, \mathbf{g}_t \in \mathbb{R}^D$, the computation of $\mathcal{L}$ requires time $\mathcal{O}(|B| \cdot (C_{\text{KL}} + C_{\text{Recon}} + C_{\text{Comp}}))$ and space $\mathcal{O}(|B| \cdot D + |B| \cdot K)$, where $C_{\text{KL}}$, $C_{\text{Recon}}$ and $C_{\text{Comp}}$ denotes the complexity of evaluating the KL divergence, reconstruction and triple completion terms in the ELBO, respectively. Please refer to Appendix A for additional proofs and computation details.

# 4 EXPERIMENT

## 4.1 EXPERIMENT SETTINGS

**Datasets**. To evaluate our method for filling in missing triples in KGs, we selected benchmark datasets ranging from moderate-sized (about 93k triples) to large-scale (around 20 million triples) for the KGC task. These include WN18RR, FB15k-237 (Toutanova et al., 2015), and Wikidata5M (Wang et al., 2021b). Originally introduced by Bordes et al. (2013), the WN18 and FB15k datasets were later refined to WN18RR and FB15k-237 following studies (Toutanova et al., 2015; Dettmers et al., 2018) that revealed test leakage issues. Textual data comes from KG-BERT (Yao et al., 2019), while Wikidata5M (Wang et al., 2021b) is a large-scale KG merging Wikidata and Wikipedia, with textual descriptions for each entity.

**Evaluation metrics**. In our approach, for each query $(h, r, ?)$, a score is calculated for each entity and the rank of the correct answer is determined. We report the Mean Reciprocal Rank (MRR) and Hit@$k$ metrics under the filtered protocol (Bordes et al., 2013). For each triple $(h, r, t)$, we construct a forward query $(h, r, ?)$ with $t$ as the answer, along with a backward query $(?, r^{-1}, t)$ for data augmentation. Here, $r^{-1}$ denotes the inverse of the relation $r$, as sourced from Li et al. (2023a). The averaged results of the forward and backward metrics are reported in our experimental evaluations.

**Baselines**. We conduct comprehensive experiments to evaluate the performance of our model against a variety of KGC models, encompassing rule-based, embedding-based and text-based KGC approaches.

**Implementation details**. To ensure a fair comparison with existing approaches, we maintain the same primary hyperparameters. Specifically, the BERT encoders are initialized with pre-trained weights from "bert-base-uncased". We use a batch size of 1024 with 4 Quadro RTX 8000 GPUs, although a larger batch size is reasonably expected to provide better performance under the contrastive framework. The maximum number of communities $K$ is consistently set to 128 for all datasets. In the case of the WN18RR and FB15k-237 datasets, we utilize in-batch negative sampling, whereas for the Wikidata5M dataset, we adopt an additional self-negative sampling strategy to ensure fair comparison with SimKGC (Wang et al., 2022).

Detailed information regarding the experimental setup can be found in Appendix B.1.

## 4.2 MAIN RESULTS

Due to the stochastic nature of our model, we perform five independent experiments with different random seeds and report the average metrics. Table 2 presents the results for the Wikidata5M dataset, while Table 1 for the WN18RR and FB15k-237 datasets. Hit@$k$ is expressed as a percentage. The best performance for each metric in each dataset is highlighted in bold, and the top metrics across categories are underlined.

Table 1: Knowledge graph completion results for the WN18RR and FB15k-237 datasets.

| Method | WN18RR | | | | FB15k-237 | | | |
|---|---|---|---|---|---|---|---|---|
| | MRR | Hit@1 | Hit@3 | Hit@10 | MRR | Hit@1 | Hit@3 | Hit@10 |
| *Rule-based Methods* | | | | | | | | |
| NeuralLP | 38.1 | 36.8 | 38.6 | 40.8 | 23.7 | 17.3 | 25.9 | 36.1 |
| DRUM | 38.2 | 36.9 | 38.8 | 41.0 | 23.8 | 17.4 | 26.1 | 36.4 |
| LERP | 62.2 | 59.3 | 63.4 | 68.2 | - | - | - | - |
| *Embedding-based Methods* | | | | | | | | |
| TransE | 24.3 | 4.3 | 44.1 | 53.2 | 27.9 | 19.8 | 37.6 | 44.1 |
| DistMult | 44.4 | 41.2 | 47.0 | 50.4 | 28.1 | 19.9 | 30.1 | 44.6 |
| R-GCN | 12.3 | 8.0 | 13.7 | 20.7 | 16.4 | 10.0 | 18.1 | 30.0 |
| RotatE | 47.6 | 42.8 | 49.2 | 57.1 | 33.8 | 24.1 | 37.5 | 53.3 |
| TuckER | 47.0 | 44.3 | 48.2 | 52.6 | 35.8 | 26.6 | 39.4 | 54.4 |
| HittER | 50.3 | 46.2 | 51.6 | 58.4 | **37.3** | **27.9** | 40.9 | **55.8** |
| N-Former | 48.6 | 44.3 | 50.1 | 57.8 | 37.2 | 27.7 | **41.2** | 55.6 |
| KRACL | 52.7 | 48.2 | 54.7 | 61.3 | 36.0 | 26.6 | 39.5 | 54.8 |
| *Text-based Methods* | | | | | | | | |
| KG-BERT | 21.6 | 4.1 | 30.2 | 52.4 | - | - | - | 42.0 |
| MTL-KGC | 33.1 | 20.3 | 38.3 | 59.7 | 26.7 | 17.2 | 29.8 | 45.8 |
| StAR | 40.1 | 24.3 | 49.1 | 70.9 | 29.6 | 20.5 | 32.2 | 48.2 |
| SimKGC | 66.6 | 58.7 | 71.7 | 80.0 | 33.6 | 24.9 | 36.2 | 51.1 |
| KG-S2S | 57.4 | 53.1 | 59.5 | 66.1 | 33.6 | 25.7 | 37.3 | 49.8 |
| GHN | 67.8 | 59.6 | 71.9 | 82.1 | 33.9 | 25.1 | 36.4 | 51.8 |
| DSLFM-KGC | **70.4** | **63.1** | **74.8** | **84.2** | 35.5 | 26.4 | 38.9 | 53.7 |

The most substantial improvement is seen on the Wikidata5M dataset, where our model shows a 5.0% increase in MRR (from 71.3% to 76.3%) and a 6.5% increase in Hit@1 (from 60.7% to 67.2%) compared to SimKGC. Similar improvements are observed on the WN18RR dataset, where DSLFM-KGC surpasses the second-best model (GHN) across all metrics, with enhancements ranging from 1.9% to 3.5% in MRR and Hit@k, demonstrating its strong predictive ca-

Table 2: KGC results for the Wikidata5M datasets.

| Method | MRR | Hit@1 | Hit@3 | Hit@10 |
|---|---|---|---|---|
| DKRL | 23.1 | 5.9 | 32.0 | 54.6 |
| KEPLER | 40.2 | 22.2 | 51.4 | 73.0 |
| BLP-ComplEx | 48.9 | 26,2 | 66.4 | 87.7 |
| BLP-SimplE | 49.3 | 28.9 | 63.9 | 86.6 |
| SimKGC | 71.3 | 60.7 | 78.7 | 91.3 |
| DSLFM-KGC | **76.3** | **67.2** | **82.7** | **93.6** |

pability. On the FB15k-237 dataset, while our model falls short of embedding-based models, it still outperforms rule-based and text-based methods, narrowing the gap between text-based and embedding-based approaches by approximately 2-3 percentage points.

To clarify the results obtained from the WN18RR and FB15k-237 datasets, we perform a detailed analysis of the underlying KGs. First, we assess the topological structure of each KG by calculating the average degree $M/N$, where $M$ and $N$ represent the number of edges and nodes, respectively. The FB15k-237 dataset exhibits a denser structure, with an average degree of 21.3, compared to 2.27 for WN18RR. Second, we examine the topological structures of both datasets. In FB15k-237, relationships show a high degree of correlation (e.g., 'award nominee', 'nominee of award'), resulting in a densely interconnected structure with a less pronounced clustering pattern. Finally, we carry out in-depth ablation studies to further examine the challenges our model experiences when capturing latent community structures from the FB15k-237 dataset, as discussed in the following section.

## 4.3 ABLATION RESULTS

We conduct diverse ablation experiments to investigate into how key components of our model impact KGC performance.

**Stick-breaking prior**. We conduct KGC experiments with $\alpha_{qry}$ and $\alpha_{ans}$ chosen from the grid $\{80, 90, 100\} \times \{10, 20, \ldots, 100\}$, while keeping all other hyperparameters fixed. Table 8 reports the mean and standard deviation of these 30 results for each dataset. The minimal variation in performance with different $\alpha_{qry}$ and $\alpha_{ans}$ values, as seen in Table 8, highlights the robustness of our model under diverse prior settings.

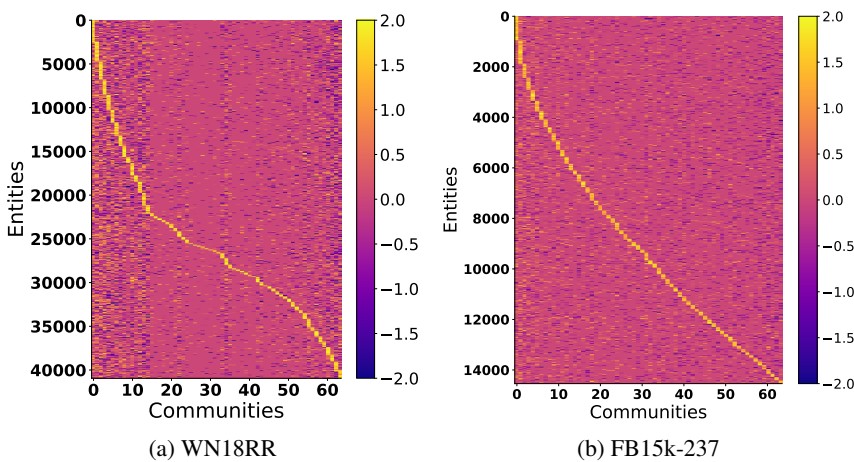

(a) WN18RR       (b) FB15k-237

Figure 4: The latent structure $F_{\text{ans}}$ learned from the WN18RR and FB15k-237 datasets. The columns of $F_{\text{ans}}$, representing communities, are sorted such that communities with higher summed strengths are assigned lower indices in the matrix.

As discussed in Section 4.2, the denser topology of the FB15k-237 dataset makes it more difficult to capture community structures. To gain further insights, we compute the average number of activated communities (the number of non-zero entries in $Z_{\text{ans}}$ divided by the number of entities $|\mathcal{E}|$) and present the trend across different $\alpha_{\text{ans}}$ values in Figure 3. Clearly, for identical $\alpha_{\text{ans}}$ values, FB15k-237 exhibits significantly fewer latent communities than WN18RR, with the disparity increasing as $\alpha_{\text{ans}}$ rises. This indicates the greater density and less pronounced clustering structure of the FB15k-237 dataset.

**Dose the sparse latent structure makes a difference**? To assess this, we replace our encoder with one that generates an approximate standard Gaussian distribution, as used in the vanilla VAE (Kingma & Welling, 2013). Additionally, we evaluate a pure autoencoder (AE), which assumes no probabilistic distribution for the latent variables. The testing performance and the training convergence epochs (based on the best validation metric) for the WN18RR and FB15k-237 datasets are shown in Table 3.

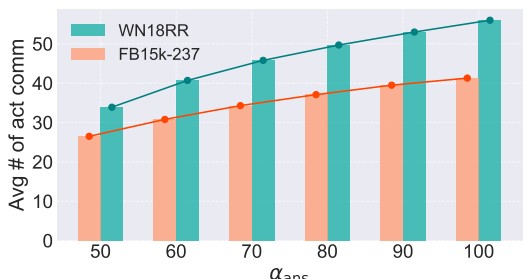

Figure 3: Average number of activated communities learned on the WN18RR and FB15k-237 datasets.

Table 3: Performance of DSLFM-KGC on the WN18RR and FB15k-237 datasets w/ different latent structures.

| Method | WN18RR | | | FB15k-237 | | |
| --- | --- | --- | --- | --- | --- | --- |
| | Hit@1 | Hit@10 | Epochs | Hit@1 | Hit@10 | Epochs |
| Ours | 63.1 | 84.2 | 65 | 26.4 | 53.7 | 15 |
| VAE | 60.9 | 82.4 | 55 | 25.6 | 52.0 | 10 |
| AE | 59.0 | 82.0 | 50 | 25.1 | 52.3 | 10 |

The results in Table 3 show that integrating latent structure substantially enhance KGC performance on the WN18RR dataset. However, the FB15k-237 dataset witnesses only modest improvements, illustrating the challenges in modeling its latent community structure. Furthermore, the increased complexity of the latent structure negatively impacted the convergence rate, as evidenced by the longer training epochs. Future studies to enhance KGC accuracy on dense-connected KGs and improve training efficiency are necessary.

## 5 ANALYSIS

To showcase the interpretability of our model, derived from SBM, we visualize the latent structure learned from the WN18RR and FB15k-237 datasets in Figure 4. For demonstration purposes, we use stick-breaking prior settings of $\alpha_{\text{qry}} = 100$, $\alpha_{\text{ans}} = 50$, and a truncation level of $K = 64$. The

Table 4: Uncovered communities from the FB15k-237 dataset along with entity descriptions. Community and entity names are highlighted in different colors, with entities in each community sorted in descending order by strength.

---

Cluster : County

County Wexford : County Wexford is a county in Ireland...

Marion County : Marion County is a county located in the U.S. state of Indiana...

County Tyrone : County Tyrone is one of the six counties of Northern Ireland...

Cluster : Music

PJ Harvey : Polly Jean Harvey MBE is an English musician...

Little Richard : ...an American recording artist, songwriter, and musician...

Italo disco : Italo disco is a genre of music which originated in Italy...

Talent manager-GB : A talent manager, also known as band manager...

---

sparse latent feature matrix $F_{ans}$ shows how entities are grouped into communities, where larger absolute values suggest stronger confidence in whether a node belongs to a specific community. For WN18RR, as illustrated in Figure 4a, more pronounced clustering is visible, with the left and right columns showing larger absolute values, while the middle columns are more moderate. In contrast, the FB15k-237 matrix exhibits more evenly distributed values across its columns.

In addition, incorporating a text encoder allows for a more in-depth understanding of the latent structure learned from a KG. We select several communities and their most significant entities from the FB15k-237 dataset for enhanced visualization, with their textual descriptions provided in Table 4. This integration of text features enables more intuitive and concrete observations of the uncovered communities, validating both the effectiveness and interpretability of our approach.

## 6 RELATED WORK

**Knowledge Graph Completion**. To address the task of KGC, initial research has concentrated on developing effective scoring mechanisms to evaluate the plausibility of triples embedded in low-dimensional spaces. A pioneering approach in this area is knowledge graph embedding (KGE) Bordes et al. (2013); Yang et al. (2014); Schlichtkrull et al. (2018); Sun et al. (2019); Balažević et al. (2019), also known as embedding-based methods. Notably, TransE Bordes et al. (2013) is a representative model that interprets a relationship $r$ as a translation from the head entity $h$ to the tail entity $t$. Tucker Balažević et al. (2019) employs Tucker Decomposition Tucker (1966) to compute a smaller core tensor and a set of three matrices, each representing entity and relation embeddings separately. Recently, text-based KGC methods have incorporated textual descriptions of entities and relations, thus encoding them into a more expressive semantic space. Specifically, NTN Socher et al. (2013) simplifies entity representation by averaging its word embeddings. SimKGC Wang et al. (2022) integrates a contrastive learning framework with three negative sampling strategies, significantly improving KGC performance. However, these prevalent KGC methods assume that the existence of a triple in a KG solely depends on the entities and relation involved, often overlooking the intricate interconnections among communities.

**KGC methods that leverage neighborhood information**. Graph Neural Networks (GNNs), especially Message Passing Neural Networks (MPNNs), have become essential tools for node representation learning in graphs, where they assume that similar neighborhood structures yield closer node representations. Notable MPNN-based KGC methods like RGCN (Schlichtkrull et al., 2018), CompGCN (Vashishth et al., 2019), and KBGAT (Nathani et al., 2019) have demonstrated strong KGC performance but have since been found to inadequately leverage neighborhood information (Zhang et al., 2022; Li et al., 2023b). Furthermore, GNN-based approaches generally do not incorporate community-level information for KG completion. Meanwhile, there are few KGC methods that leverage clustering features, such as CTransR (Lin et al., 2015) and EL-Trans (Yang et al., 2023).

However, these models often struggle with poor KGC performance and lack an end-to-end design, limiting their applicability to modern KGs.

**Stochastic Blockmodels** have demonstrated success in uncovering various latent structures, thereby enhancing link prediction. The stochastic blockmodel (SBM) Holland et al. (1983) assigns each node to a specific community, with the interconnections between nodes influenced by their community memberships. The mixed membership stochastic blockmodel (MMSB) Airoldi et al. (2008) introduces a multinomial indicator vector for node-community assignments, allowing for mixed membership communities. However, MMSB restricts nodes to a single cluster at any given time. The overlapping stochastic blockmodel (OSBM) Latouche et al. (2011) overcomes this limitation by utilizing a multi-Bernoulli distribution, enabling nodes to belong to multiple communities simultaneously. The latent feature relational model (LFRM) Miller et al. (2009) is a specific instance of OSBM that applies the Indian Buffet Process (IBP) prior to the assignment matrix. Traditional SBMs, however, are constrained in expressiveness and scalability due to their reliance on MCMC Miller et al. (2009) or variational inference Zhu et al. (2016) for learning latent variables. Recently, DGLFRM Mehta et al. (2019) employs a sparse variational autoencoder (VAE) framework for inference in SBMs, thereby extending their applicability to larger graphs. Nevertheless, DGLFRM struggles to handle graphs with tens of thousands of nodes or more, a common scenario in modern KGs.

# 7 CONCLUSION

In this paper, we propose DSLFM-KGC, a framework developed to learn sparse latent structural features for enhancing knowledge graph completion (KGC). Specifically, we introduce a novel generative model for KGs, based on stochastic blockmodels (SBMs), which dynamically uncover latent communities and improve triple completion performance. Additionally, a deep sparse variational autoencoder enables scalable inference and greater expressiveness. Extensive experiments on three benchmark datasets verify the superior performance of DSLFM-KGC in completing missing triples while maintaining interpretability. Despite the improvements in KGC performance, there is still a significant challenge in optimizing training efficiency. Future research will focus on learning more expressive latent representations while reducing computational overhead.

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

# A  MATHEMATICAL PROOFS

This section provides detailed mathematical derivations of the negative ELBO as introduced in Equation 17.

## A.1  THE NEGATIVE ELBO

Let $\mathcal{H} = \{V_{\text{qry}}, Z_{\text{qry}}, W_{\text{qry}}, V_{\text{ans}}, Z_{\text{ans}}, W_{\text{ans}}\}$ denote the set of latent variables and $\mathcal{O} = \{X_{\text{qry}}, X_{\text{ans}}, A\}$ the set of observations, with $X_{\text{qry}}$ and $X_{\text{ans}}$ being the tokenized sequences of the queries and answer, respectively. The negative ELBO in our model is formulated as:

$$
\begin{aligned}
\mathcal{L} &= -\mathbb{E}_q \left[ \log \frac{p_\theta(\mathcal{H}, \mathcal{O})}{q_\phi(\mathcal{H})} \right] \\
&= -\mathbb{E}_q \left[ \log \frac{p_\theta(\mathcal{H})}{q_\phi(\mathcal{H})} + \log p_\theta(\mathcal{O}|\mathcal{H}) \right] \\
&= D_{\text{KL}} \left[ q_\phi(\mathcal{H}) || p_\theta(\mathcal{H}) \right] - \mathbb{E}_q \left[ \log p_\theta(X_{\text{qry}}, X_{\text{ans}}, A | Z_{\text{qry}}, Z_{\text{ans}}, W_{\text{qry}}, W_{\text{ans}}) \right] \\
&= D_{\text{KL}} \left[ q_\phi(\mathcal{H}) || p_\theta(\mathcal{H}) \right] - \mathbb{E}_q \left[ \log p_\theta(X_{\text{qry}} | Z_{\text{qry}}, W_{\text{qry}}) \right] \\
&\quad - \mathbb{E}_q \left[ \log p_\theta(X_{\text{qry}} | Z_{\text{qry}}, W_{\text{qry}}) \right] - \mathbb{E}_q \left[ \log p_\theta(X_{\text{ans}} | Z_{\text{ans}}, W_{\text{ans}}) \right] \\
&\quad - \mathbb{E}_q \left[ \log p_\theta(A | Z_{\text{qry}}, Z_{\text{ans}}, W_{\text{qry}}, W_{\text{ans}}) \right]
\end{aligned}
$$

This objective consists of three main parts: the first two KL divergence terms $\mathcal{L}_{\text{KL}}$, the next two reconstruction terms $\mathcal{L}_{\text{recon}}$, and the last triple completion term $\mathcal{L}_{\text{comp}}$.

Given a batch of triples $B \subset \mathcal{G}$, with $\mathcal{Q}_B$ and $\mathcal{E}_B$ representing the associated queries and answers, we derive a batch-optimized version of Equation 17:

$$
\begin{aligned}
\mathcal{L}(B) &= \mathcal{L}_{\text{KL}}(B) + \mathcal{L}_{\text{Recon}}(B) + \mathcal{L}_{\text{Comp}}(B) \\
&= \sum_{(h,r) \in \mathcal{Q}_B} \left\{ D_{\text{KL}} \left[ q_\phi(\mathbf{v}_{hr}) || p_\theta(\mathbf{v}_{hr}) \right] + D_{\text{KL}} \left[ q_\phi(\mathbf{z}_{hr}) || p_\theta(\mathbf{z}_{hr} | \mathbf{v}_{hr}) \right] + D_{\text{KL}} \left[ q_\phi(\mathbf{w}_{hr}) || p_\theta(\mathbf{w}_{hr}) \right] \right\} \\
&\quad + \sum_{t \in \mathcal{E}_B} \left\{ D_{\text{KL}} \left[ q_\phi(\mathbf{v}_t) || p_\theta(\mathbf{v}_t) \right] + D_{\text{KL}} \left[ q_\phi(\mathbf{z}_t) || p_\theta(\mathbf{z}_t | \mathbf{v}_t) \right] + D_{\text{KL}} \left[ q_\phi(\mathbf{w}_t) || p_\theta(\mathbf{w}_t) \right] \right\} \\
&\quad - \sum_{(h,r) \in \mathcal{Q}_B} \mathbb{E}_q \left[ \log p_\theta(\mathbf{x}_{hr} | \mathbf{z}_{hr}, \mathbf{w}_{hr}) \right] - \sum_{t \in \mathcal{E}_B} \mathbb{E}_q \left[ \log p_\theta(\mathbf{x}_t | \mathbf{z}_t, \mathbf{w}_t) \right] \\
&\quad - \sum_{(h,r,t) \in B} \mathbb{E}_q \left[ \log p_\theta(A_{hr,t} | \mathbf{z}_{hr}, \mathbf{z}_t, \mathbf{w}_{hr}, \mathbf{w}_t) \right]
\end{aligned} \tag{19}
$$

To compute the reconstruction terms, such as $\mathbb{E}_q \left[ \log p_\theta(\mathbf{x}_{hr} | \mathbf{z}_{hr}, \mathbf{w}_{hr}) \right]$, we use the cosine similarity between the embedding $\mathbf{e}_{hr}$ (Equation 14) and the decoded representation $\mathbf{g}_{hr}$ (Equation 16):

$$
\mathbb{E}_q \left[ \log p_\theta(\mathbf{x}_{hr} | \mathbf{z}_{hr}, \mathbf{w}_{hr}) \right] = \cos(\mathbf{e}_{hr}, \mathbf{g}_{hr}) \tag{20}
$$

Note that, $\mathbf{v}_{hr}, \mathbf{v}_t, \mathbf{z}_{hr}, \mathbf{z}_t, \mathbf{w}_{hr}$ and are $K$-dimensional vectors, while $\mathbf{g}_{hr}, \mathbf{g}_t \in \mathbb{R}^D$. The time required to compute $\mathcal{L}_{\text{KL}}(B)$, $\mathcal{L}_{\text{Recon}}(B)$ and $\mathcal{L}_{\text{Comp}}(B)$ in Equation 19 is $\mathcal{O}(|B| \cdot C_{\text{KL}})$, $\mathcal{O}(|B| \cdot C_{\text{Recon}})$ and $\mathcal{O}(|B| \cdot C_{\text{Comp}})$, with space complexities $\mathcal{O}(|B| \cdot K)$, $\mathcal{O}(|B| \cdot D)$ and $\mathcal{O}(|B| \cdot D)$, respectively. Here, $C_{\text{KL}}$, $C_{\text{Recon}}$ and $C_{\text{Comp}}$ denote the complexity for evaluating the KL divergence, reconstruction and triple completion terms for a single triple. Thus, the total time and space complexity for computing 19 are $\mathcal{O}(|B| \cdot (C_{\text{KL}} + C_{\text{Recon}} + C_{\text{Comp}}))$ and $\mathcal{O}(|B| \cdot D + |B| \cdot K)$.

In practice, we apply two different weighting coefficients to the KL and reconstruction losses to balance the learning objectives and reduce the risk of posterior collapse (Higgins et al., 2017):

$$
\mathcal{L}(B) = \beta \mathcal{L}_{\text{KL}}(B) + \eta \mathcal{L}_{\text{Recon}}(B) + \mathcal{L}_{\text{Comp}}(B) \tag{21}
$$

Regarding the KL terms, we adhere to the method described by Kingma & Welling (2013) for computing the KL divergences for two normal variables: $D_{\text{KL}} \left[ q_\phi(\mathbf{w}_{hr}) || p_\theta(\mathbf{w}_{hr}) \right]$ and $D_{\text{KL}} \left[ q_\phi(\mathbf{w}_t) || p_\theta(\mathbf{w}_t) \right]$. In the following sections, we derive the computation of the KL divergence for two Beta distributions, i.e., $D_{\text{KL}} \left[ q_\phi(\mathbf{v}_{hr}) || p_\theta(\mathbf{v}_{hr}) \right]$ and $D_{\text{KL}} \left[ q_\phi(\mathbf{v}_t) || p_\theta(\mathbf{v}_t) \right]$, as well as for two Bernoulli distributions, i.e., $D_{\text{KL}} \left[ q_\phi(\mathbf{z}_{hr}) || p_\theta(\mathbf{z}_{hr} | \mathbf{v}_{hr}) \right]$ and $D_{\text{KL}} \left[ q_\phi(\mathbf{z}_t) || p_\theta(\mathbf{z}_t | \mathbf{v}_t) \right]$.

## A.2   The KL divergence of Beta distributions

The KL divergence of two Beta distributions has a closed-form solution. The PDF of a Beta distribution Beta$(a, b)$ with concentration parameters $a, b$ is given by:

$$f(x|a, b) = \frac{1}{\mathrm{B}(a, b)} x^{a-1}(1 - x)^{b-1}, \quad 0 \leq x \leq 1 \tag{22}$$

where $\mathrm{B}(a, b)$ is the Beta function, defined as

$$\mathrm{B}(a, b) = \int_0^1 u^{a-1}(1 - u)^{b-1} du = \frac{\Gamma(a)\Gamma(b)}{\Gamma(a + b)} \tag{23}$$

with $\Gamma(\cdot)$ representing the Gamma function.

Let the distributions $p(x)$ and $q(x)$ be Beta$(a_p, b_p)$ and Beta$(a_q, b_q)$ respectively, the KL divergence for $q(x)$ and $p(x)$ is computed as:

$$
\begin{aligned}
KL\left[q(x)||p(x)\right] &= \mathbb{E}_q\left[\log \frac{q(x)}{p(x)}\right] \\
&= \mathbb{E}_q\left[\log \frac{\frac{1}{\mathrm{B}(a_q, b_q)} x^{a_q-1}(1 - x)^{b_q-1}}{\frac{1}{\mathrm{B}(a_p, b_p)} x^{a_p-1}(1 - x)^{b_p-1})}\right] \\
&= \mathbb{E}_q\left[\log \frac{\mathrm{B}(a_p, b_p)}{\mathrm{B}(a_q, b_q)}\right] + (a_q - a_p)\mathbb{E}_q\left[\log x\right] + (b_q - b_p)\mathbb{E}_q\left[\log(1 - x)\right] \\
&= \log \mathrm{B}(a_p, b_p) - \log \mathrm{B}(a_q, b_q) + (a_q - a_p)\mathbb{E}_q\left[\log x\right] + (b_q - b_p)\mathbb{E}_q\left[\log(1 - x)\right]
\end{aligned}
$$

where $\mathbb{E}_q[\log x]$ and $\mathbb{E}_q[\log(1 - x)]$ are the expected sufficient statistics under distribution $q$, which can be computed using the properties of the exponential family distributions:

$$\mathbb{E}_q[\log x] = \psi(a_q) - \psi(a_q + b_q) \tag{24}$$
$$\mathbb{E}_q[\log(1 - x)] = \psi(b_q) - \psi(a_q + b_q) \tag{25}$$

where $\psi(\cdot)$ denotes the di-gamma function.

Thus, the complete expression of the KL divergence becomes:

$$
\begin{aligned}
KL\left[q(x)||p(x)\right] = \log \mathrm{B}(a_p, b_p) - \log \mathrm{B}(a_q, b_q) + (a_q - a_p)(\psi(a_q) - \psi(a_q + b_q)) \\
+ (b_q - b_p)(\psi(b_q) - \psi(a_q + b_q))
\end{aligned} \tag{26}
$$

## A.3   The KL divergence of Concrete distributions

To enable differentiable optimization, we utilize the binary Concrete distribution (Maddison et al., 2016) to obtain a continuous relaxation of the Bernoulli distribution (Equation 12). However, the KL divergence of two Concrete distributions, $q(y)$ and $p(y)$, is intractable. We resort to approximation using the Monte Carlo (MC) expectations:

$$
\begin{aligned}
KL\left[q(y)||p(y)\right] &= \mathbb{E}_q\left[\log q(y) - \log p(y)\right] \\
&\simeq \frac{1}{N}\sum_{i=1}^N (\log q(y_i) - \log p(y_i)), \quad y_i \sim q(y), \ i = 1, \dots, N
\end{aligned} \tag{27}
$$

According to Maddison et al. (2016), the logarithm of the probability density function for the Concrete distribution is given by:

$$\log p(y|\pi, \lambda) = \log \lambda - \lambda y + \log \pi - 2\log(1 + \exp(-\lambda y + \log \pi)) \tag{28}$$

where $p(y|\pi, \lambda) \triangleq \mathrm{Concrete}(y|\pi, \lambda)$ denotes the Concrete distribution, $\lambda \in (0, \infty)$ the relaxation temperature, and $\pi$, the probability ratio.

Table 5: Statistics of datasets.

| Dataset | # Relation | # Entity | # Triple | # Train | # Validation | # Test |
|---------|-----------|----------|----------|---------|--------------|--------|
| WN18RR | 11 | 40,943 | 93,003 | 86,835 | 3,034 | 3,134 |
| FB15k-237 | 237 | 14,541 | 310,116 | 272,115 | 17,535 | 20,466 |
| Wikidata5M | 822 | 4,594,485 | 20,624,605 | 20,614,279 | 5,163 | 5,163 |

Specifically, the KL divergence term $D_{\mathrm{KL}}\left[q_\phi(\mathbf{z}_{hr})||p_\theta(\mathbf{z}_{hr}|\mathbf{v}_{hr})\right]$ in the negative ELBO (Equation 17) is computed as:

$$D_{\mathrm{KL}}\left[q_\phi(\mathbf{z}_{hr})||p_\theta(\mathbf{z}_{hr}|\mathbf{v}_{hr})\right] = \mathbb{E}_q\left[\log q_\phi(\mathbf{z}_{hr}) - \log p_\theta(\mathbf{z}_{hr}|\mathbf{v}_{hr})\right]$$

$$= \sum_{k=1}^{K}\mathbb{E}_q[\log q_\phi(z_{hr,k}) - \log p_\theta(z_{hr,k}|\mathbf{v}_{hr})] \tag{29}$$

where we apply the Concrete relaxation to the variational posterior (Equation 12) and the prior (Equation 9):

$$q_\phi(z_{hr,k}) \triangleq \mathrm{Concrete}(z_{hr,k}|\pi_{hr,k}(\mathcal{G}), \lambda_{\mathrm{post}}) \tag{30}$$

$$p_\theta(z_{hr,k}|\mathbf{v}_{hr}) \triangleq \mathrm{Concrete}(z_{hr,k}|\pi_{hr,k}(\mathbf{v}_{hr}), \lambda_{\mathrm{prior}}) \tag{31}$$

In this case, $\lambda_{\mathrm{post}}$ and $\lambda_{\mathrm{prior}}$ are hyperparameters and we have

$$\pi_{hr,k}(\mathbf{v}_{hr}) = \prod_{j=1}^{k} v_{hr,j}, \ v_{hr,j} \sim q_\phi(v_{hr,j}) \tag{32}$$

where $q_\phi(v_{hr,j})$ is defined in Equation 11. Then $D_{\mathrm{KL}}\left[q_\phi(\mathbf{z}_{hr})||p_\theta(\mathbf{z}_{hr}|\mathbf{v}_{hr})\right]$ is estimated using Equation 27. The computation of $D_{\mathrm{KL}}\left[q_\phi(\mathbf{z}_{hr})||p_\theta(\mathbf{z}_{hr}|\mathbf{v}_{hr})\right]$ is implemented similarly.

### A.4 REPARAMETERIZATION

In our model, the expectations over Beta, Bernoulli and Normal distributions is approximated using differentiable Monte Carlo (MC) estimate, as required by SGVB (Kingma & Welling, 2013). Furthermore, to draw samples from these distributions, a reparameterization trick is needed to ensure effective differentiation. To sample Normal variables $\mathbf{w}_{hr}$ and $\mathbf{w}_t$, we follow the standard approach used in vanilla VAE (Kingma & Welling, 2013). For reparameterization of Beta variables $\mathbf{v}_{hr}$ and $\mathbf{v}_t$, we adopt the implicit differentiation method (Figurnov et al., 2018).

To draw discrete Bernoulli variables during training, we utilize the Gumble-max relaxation (Maddison et al., 2014; Jang et al., 2016) to achieve a continuous approximation. Specifically, the distribution used to reparameterize $z_{hr,k}$ aligns with a binary special case of the Concrete distribution (Maddison et al., 2016):

$$u \sim \mathrm{Uniform}(0,1) \quad L = \log(u) - \log(1-u)$$

$$y_{hr,k} \overset{d}{=} (\mathrm{logit}(\pi_{hr,k}) + L)/\lambda$$

$$z_{hr,k} = \sigma\left(y_{hr,k}\right) \tag{33}$$

where $z_{hr,k}$ and $\pi_{hr,k}$ are defined in Equation 12 and 15, $\mathrm{logit}(\cdot)$ is the inverse-sigmoid function and $\lambda$ is the relaxation temperature. The reparameterization of $z_{t,k}$ is achieved similarly.

## B EXPERIMENT

### B.1 EXPERIMENT SETTINGS

**Datasets**. The statistics of each dataset are shown in Table 5.

**Baselines**. The baseline methods we choose can be categorized into three classes:

Table 6: Hyperparameters of our DSLFM-KGC for each dataset during training.

| Hyperparameter | WN18RR | FB15k-237 | Wikidata5M |
|---|---|---|---|
| initial learning rate | $8 \times 10^{-5}$ | $2 \times 10^{-5}$ | $5 \times 10^{-5}$ |
| epochs | 65 | 15 | 1 |
| contrastive temperature $\tau$ | 0.02 | 0.08 | 0.03 |
| dropout | 0 | 0.1 | 0 |
| stick-breaking prior $\alpha_{\mathrm{qry}}$ | 100 | 100 | 100 |
| stick-breaking prior $\alpha_{\mathrm{ans}}$ | 20 | 20 | 100 |
| truncation level $K$ | 128 | 128 | 128 |

Table 7: The parameter count, training epochs, and GPU hours required by SimKGC (Wang et al., 2022) and DSLFM-KGC.

| Model | # Params | WN18RR | | FB15k-237 | | Wikidata5M | |
|---|---|---|---|---|---|---|---|
| | | Epochs | GPU hours | Epochs | GPU hours | Epochs | GPU hours |
| SimKGC | 218.0M | 50 | 3 | 10 | 2 | 1 | 12 |
| DSLFM-KGC (ours) | 219.8M | 65 | 3.5 | 15 | 3 | 1 | 13 |

- For rule-based methods, we incorporate NeuralLP (Yang et al., 2017), DRUM (Sadeghian et al., 2019) and LERP (Han et al., 2023).

- In the category of embedding-based methods, we choose TransE (Bordes et al., 2013), DistMult (Yang et al., 2014), R-GCN (Schlichtkrull et al., 2018), ConvE (Dettmers et al., 2018), RotatE (Sun et al., 2019), TuckER (Balažević et al., 2019), HittER (Chen et al., 2021), N-Former (Liu et al., 2022) and KRACL Tan et al. (2023).

- Text-based methods considered include KG-BERT (Yao et al., 2019), MTL-KGC (Kim et al., 2020), StAR (Wang et al., 2021a), KG-S2S (Chen et al., 2022), DKPL (Xie et al., 2016), KEPLER (Wang et al., 2021b), BLP Daza et al. (2021), SimKGC (Wang et al., 2022) and GHN (Qiao et al., 2023).

**Implementation details**. We utilize two separate BERT encoders to process the textual descriptions of the queries and answers. For a specific query $(h, r)$ and entity $t$, the token sequences, *i.e.*, $\mathbf{x}_{hr}$ and $\mathbf{x}_t$, are defined as follows:

$$x_{hr} = [\mathrm{CLS}, \mathcal{M}(h), \mathrm{SEP}, \mathcal{M}(r), \mathrm{SEP}] \tag{34}$$
$$x_t = [\mathrm{CLS}, \mathcal{M}(t), \mathrm{SEP}] \tag{35}$$

where CLS and SEP are special tokens introduced by Devlin et al. (2019), and $\mathcal{M}(h), \mathcal{M}(r)$, and $\mathcal{M}(t)$ represent the tokenized textual descriptions of the head, relation and tail, respectively. Following tokenization, $\mathbf{x}_{hr}$ and $\mathbf{x}_t$ are processed through BERT encoders, as specified in Equation 14.

**Hyerparameter**. Table 6 lists the consistent hyperparameters used for each dataset.

### B.2 ADDITIONAL ABLATION RESULTS

Figure 5 and Table 9 depict the training behavior and testing performance of DSLFM-KGC across various KL weight $\beta$ settings. For both the WN18RR and FB15k-237 datasets, setting $\beta = 10^{-1}$ leads to a learning imbalance between the KL and triple completion losses, which negatively impacts the validation loss. In contrast, the validation loss ($\mathcal{L}_{\mathrm{comp}}$) curves for DSLFM-KGC with $\beta = 10^{-2}, 10^{-3}$, and $10^{-4}$ show minimal variation. This observation is mirrored in the testing results shown in Table 9.

Table 8: Performance of DSLFM-KGC on the WN18RR, FB15k-237 and Wikidata5M datasets w/ different stick-breaking priors.

| Dataset | WN18RR | | FB15k-237 | | Wikidata5M | |
|---|---|---|---|---|---|---|
| | Hit@1 | Hit@10 | Hit@1 | Hit@10 | Hit@1 | Hit@10 |
| Mean | 62.7 | 84.1 | 26.2 | 53.8 | 66.9 | 94.0 |
| Std | 0.3 | 0.1 | 0.2 | 0.1 | 0.4 | 0.2 |

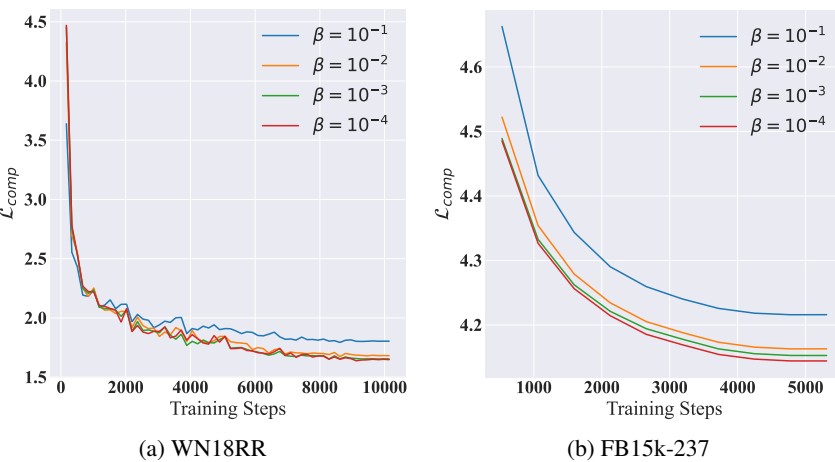

(a) WN18RR        (b) FB15k-237

Figure 5: Validation triple completion loss $\mathcal{L}_{\text{comp}}$ for DSLFM-KGC during training with different $\beta$ values on the WN18RR and FB15k-237 datasets.

Table 9: Performance of DSLFM-KGC on the WN18RR and FB15k-237 datasets w/ different $\beta$ values.

| $\beta$ | WN18RR | | | FB15k-237 | | |
|---|---|---|---|---|---|---|
| | MRR | Hit@1 | Hit@10 | MRR | Hit@1 | Hit@10 |
| $10^{-1}$ | 69.2 | 61.6 | 83.3 | 33.7 | 24.5 | 52.2 |
| $10^{-2}$ | 70.2 | 62.8 | 83.9 | 35.1 | 26.0 | 53.3 |
| $10^{-3}$ | 70.2 | 62.6 | 84.3 | 35.4 | 26.2 | 53.6 |
| $10^{-4}$ | 70.4 | 62.5 | 84.0 | 35.4 | 26.2 | 53.7 |

## C RELATED WORK

**KGC with large language models (LLMs).** Recent advancements in text-based KGC leverage the extensive pre-trained knowledge and contextual understanding of LLMs to bridge the gap between structured and unstructured knowledge. Techniques in this domain often employ diverse prompt designs to enable LLMs to perform direct reasoning for KGC (Yao et al., 2023; Wei et al., 2024) or to refine textual information in datasets, enhancing their accuracy and richness (Li et al., 2023a; Yang et al., 2024). However, while these methods are training-free and inherently interpretable, they face challenges such as hallucinations and reliance on few-shot demonstrations, which are difficult to implement in sparsely connected KGs like WN18RR. Alternatively, some approaches fine-tune LLMs on KGC tasks using strategies like prefix-tuning (Chen et al., 2023; Zhang et al., 2024b) or adapter-tuning. While these methods capitalize on the reasoning capabilities of LLMs, they often lack interpretability, struggle to generalize across datasets, and continue to face challenges in achieving strong performance. In contrast, our model excels on relatively sparse KGs with distinct clustering patterns, leveraging text not only to improve KGC interpretability but also to provide meaningful clustering information about the KG itself. Additionally, while LLMs provide external knowledge

to enhance KGC, our approach focuses on directly extracting and utilizing the intrinsic information within KGs to strengthen representation learning. This makes our method particularly effective in scenarios where LLMs cannot reliably provide external knowledge, such as in domain-specific datasets.

Our work also relates closely to **Variational AutoEncoders (VAEs)** (Kingma & Welling, 2013), a foundational class of generative models that employs an encoder to map input data to a latent space, typically assuming a Gaussian prior, and a decoder to reconstruct the data from this latent representation. To facilitate gradient-based optimization during training, the reparameterization trick is used, re-expressing the sampling of latent variables as deterministic functions of noise variables, thereby enabling backpropagation through stochastic nodes. While this trick is straightforward for "location-scale" distributions like the Gaussian, extending it to other distributions such as Bernoulli (Jang et al., 2016; Maddison et al., 2016) and Beta distributions (Nalisnick & Smyth, 2016) requires more sophisticated techniques. Reparameterization for these distributions often involves implicit differentiation methods to compute gradients when explicit reparameterization is infeasible (Figurnov et al., 2018). A persistent challenge in training VAEs is posterior collapse, where the encoder's output becomes similar to the prior, causing the model to ignore the latent variables (Bowman et al., 2015). This issue undermines the VAE's ability to learn meaningful representations. Various strategies have been proposed to mitigate posterior collapse, including modifying the objective function with $\beta$VAE to balance reconstruction and regularization terms (Higgins et al., 2017), employing annealing schedules for the KL divergence term (Bowman et al., 2015), and designing more expressive posterior distributions to better capture the underlying data structure (Rezende & Mohamed, 2015).

