# OpenReview forum: "Deep Sparse Latent Feature Models for Knowledge Graph Completion"
_ICLR.cc/2025/Conference — Submitted to ICLR 2025_

### Official Review · Reviewer_2RBB · 2024-10-26

**Soundness:** 3
**Presentation:** 3
**Contribution:** 2
**Rating:** 5
**Confidence:** 4

**Summary:**

This paper proposes a sparse latent feature model that is optimized via a deep variational autoencoder to achieve interpretable completions for knowledge graphs. Experimental results verify the effectiveness of the proposed model.

**Strengths:**

1. This paper is well-written and easy to follow
2. Many experiments are conducted to demonstrate the effectiveness of the proposed method.

**Weaknesses:**

1. My biggest concern is the motivation of the proposed method. The authors claim that existing text-based methods overlook the complex interconnectivity among entities, however, there are some GNN-based TKGC models such as SEA-KGC [1] that can model graph information, what's the superiority of the proposed method compared with GNN-based methods? Moreover, given the text descriptions of the entities, many text-attributed graph embedding methods can also be applied to complete knowledge graphs, such as GLEM [2] and GraphFormers [3]. It's interesting to see their performance and further analysis of is there any unique challenges of applying these TAG models for text-based KGC.

2. Recently some methods have proposed to use LLMs to understand the text semantics within TKGs to perform completion or reasoning tasks (e.g., CSProm-KG [4] and KICGPT [5]), which have gained good performance and generalization ability across different KGs. The authors should provide more comparison and analysis for these methods since they also use text information within KGs.


[1] Unifying Structure and Language Semantic for Efficient Contrastive Knowledge Graph Completion with Structured Entity Anchors

[2] Learning on Large-scale Text-attributed Graphs via Variational Inference

[3] GraphFormers: GNN-nested transformers for representation learning on textual graph

[4] Dipping PLMs Sauce: Bridging Structure and Text for Effective Knowledge Graph Completion via Conditional Soft Prompting

[5] KICGPT: Large Language Model with Knowledge in Context for Knowledge Graph Completion

**Questions:**

See weakness

---

> ### Author Response · Authors · 2024-11-15
> **Clarifying Model Motivation, Comparisons with GNN and TAG Models, and Baseline Selection for Text-Based KGC**
>
> Thank you for the reviewer’s thoughtful feedback and valuable suggestions regarding motivation, clarity, and baseline selection. Below, we address each of the reviewer’s comments individually, including references to the specific revisions made in the manuscript.
>
> ## Response to Weakness 1
> Thank you for your insightful feedback regarding the motivation and positioning of our proposed method relative to GNN-based and TAG models. We would like to provide a detailed comparison between our method and each of the mentioned type of models.
>
> 1. Regarding the difference between GNN-based methods and our approach in integrating complex connectivity information among nodes, our approach consider **not only the features of each triple itself** when determining relationships between entities, **but also the underlying clustering information**. As illustrated in Figure 1, if there is a strong connectivity between two communities, it is likely that nodes in these communities may also share certain relationships.
>
> - GNN-based methods typically assume that **nodes with similar neighborhoods will have similar representations for learning node embeddings**. This **contrasts sharply** with our approach, which addresses KGC through **relationships between clusters**. In addition, SEA-KGC integrates pretrained language models with KGE methods to capture semantic structure **without relying on GNNs**. Studies [1][2] further indicate that **Message Passing GNN-based KGC methods struggle to effectively leverage neighborhood information**.
>
> - To clarify these points, we have **revised the manuscript**, specifically updating the **motivation** in the Introduction (Line 41, Page 1), expanding our **listed contributions** (Line 84, Page 2), and adding a dedicated section in **Related Work** comparing KGC methods that **leverage neighborhood information** (Line 479, Page 9).
>
> 2. GLEM and GraphFormers both utilize GNNs and LMs to integrate graph structure information with node text information. Below, we discuss these two models separately:
>
> - **GLEM** effectively combines global information extracted by GNNs with local information by LMs through an EM training approach, **targeting node classification tasks**. Its **stepwise extraction and integration of multimodal features via the EM algorithm** provide a valuable strategy.
>
> - **GraphFormers** are essentially a type of representation encoder for textual graphs that combines GNNs and transformers, which could directly replace the Bert encoder we currently use to generate more expressive representations, with an anticipated improvement in KGC performance. However, **for fair comparisons, most text-based models currently use bert-base-uncased during evaluation**.
>
> In general, there are several challenges when applying TAG models to text-based KGC:
> - Effectively incorporating additional edge types (relations) into the model.
> - As noted above, the GNN module needs careful design to fully utilize neighborhood features.
> - Text-based KGC often relies on large batch sizes for robust contrastive training, where traditional MLE training typically falls short. This necessitates lightweight implementations of models like GraphFormers to accommodate these requirements.
>
> ## Response to Weakness 2
> Certainly, recent approaches have achieved notable success in KGC tasks by leveraging LLMs. These methods either directly utilize the **strong reasoning capabilities of LLMs to extract triple features** (CSProm-KG) or employ them to **generate additional information that enriches existing knowledge** (KICGPT). In contrast, our work focuses on **enhancing representation learning** of existing text-based KGC methods to achieve **better performance and improved interpretability**. We believe that directly comparing two types of models towards fundamentally different optimization direction, plus vastly different parameter scales, may lead to unfair evaluations.
>
> While we have not included some LLM-based baselines for comparison, we have also incorporated several recent state-of-the-art methods from both embedding-based (N-Former, KRACL) and text-based (KG-S2S, GHN) approaches to ensure a comprehensive evaluation in Table 1, Page 7.
>
> [1] Are Message Passing Neural Networks Really Helpful for Knowledge Graph Completion?
>
> [2] Rethinking graph convolutional networks in knowledge graph completion.

---

> > ### Comment · Reviewer_2RBB · 2024-11-19
> >
> > Thanks to the authors for their feedback.
> > For weakness 1, I understand that the proposed method has considered the clustering information that the simple message-passing mechanism can not capture. However, more direct intuition examples should be provided to clarify why clustering information is important for TKGC. In what cases can the message-passing mechanism fail, but the proposed model succeeds? Also, I believe including typical GNN-based TKGC models as baselines helps to demonstrate the superiority of the proposed model.
> >
> > For weakness 2, I still think it's necessary to analyze the advantages of the proposed model compared with LLM-based models (such as LLM suffer from hallucinations), and thus clarify in what special case we should use the proposed model rather than an LLM-based model (given that LLM is powerful and training-free). Moreover, the authors emphasize the interpretability (latent structure) of the proposed model, but LLM can provide more human-readable Interpretations of the predictions. Although a direct comparison of the completion performance of LLM-based models is not required, a more detailed discussion is necessary.

---

> > > ### Author Response · Authors · 2024-11-22
> > > **Clarifying Comparisons with GNN-Based and LLM-Based TKGC Methods**
> > >
> > > ## Comparison with GNN-based TKGC methods
> > > Thank you for your suggestion. We would like to provide further clarification on why GNN-based TKGC methods were not included in our comparisons.
> > >
> > > **Why message passing fails in KGC**. Studies [1][2] have shown that Message Passing GNN-based KGC methods struggle to effectively utilize neighborhood information, as detailed below:
> > >
> > > - In the work of [1], they conduct experiment on WN18RR and FB15k-237 datasets with various GNNs, including RGCN, WGCN and CompGCN, where the **edges are all eliminated**. The results demonstrate that these edge-removed models perform competitively with their original counterparts, indicating that **performance gains are not derived from neighborhood aggregation**.
> > >
> > > - Similarly, [2] replaces GNNs with a simple MLP, achieving nearly identical performance. The results further suggest that the message-passing mechanism contributes minimally to KGC performance.
> > >
> > > In addition, despite an extensive literature review, we found **very few TKGC methods that leverage GNNs**.
> > >
> > > Based on these observations and supporting evidence, we have not included many GNN-based KGC methods in our comparisons. To clarify these points, we have revised the manuscript, adding a dedicated section in Related Work comparing KGC methods that leverage neighborhood information (Line 479, Page 9).
> > >
> > > ## Comparison with LLM-based TKGC methods
> > > Thank you for your valuable feedback on this point. We have **revised the Related Work section (Line 1014, Page 19)** to include a more detailed discussion of the differences between our method and those based on LLMs.
> > >
> > > - In summary, while LLM-based methods are training-free and inherently interpretable, they face notable challenges such as hallucinations and a reliance on few-shot demonstrations, which are difficult to apply to sparsely connected KGs like WN18RR. In contrast, our model excels on such sparse KGs with distinct clustering patterns by leveraging text to enhance KGC interpretability and provide meaningful clustering insights about the KG itself.
> > >
> > > - Additionally, LLM-based methods rely on external knowledge to boost KGC performance, whereas our approach focuses on extracting and utilizing the rich, intrinsic information within KGs to strengthen representation learning. This makes our method particularly effective in scenarios where LLMs cannot reliably provide external knowledge, such as in domain-specific datasets.
> > >
> > > [1] Are Message Passing Neural Networks Really Helpful for Knowledge Graph Completion?
> > >
> > > [2] Rethinking graph convolutional networks in knowledge graph completion.

---

> > > > ### Author Response · Authors · 2024-11-29
> > > > **Gentle Reminder: Final Discussion Stage**
> > > >
> > > > Dear Reviewer 2RBB,
> > > >
> > > > We sincerely appreciate your thoughtful comments, which have significantly contributed to improving our paper. As the discussion phase nears its conclusion, we wanted to send a gentle reminder to ask if you might have any additional questions or points of clarification regarding our submission. We are happy to address them while there is still time.

---

### Official Review · Reviewer_8T3f · 2024-10-31

**Soundness:** 2
**Presentation:** 2
**Contribution:** 2
**Rating:** 5
**Confidence:** 5

**Summary:**

This paper proposes a novel deep sparse latent feature model for knowledge graph completion. It is optimized through VAE, with the goal of uncovering the latent structures in KGs and completing missing triples. Experiments demonstrate that this approach outperforms existing methods in terms of both performance and interpretability.

**Strengths:**

Strengths
1. Experimental Results: The model shows significant performance improvements on the Wikidata5M dataset (e.g., a 5% increase in MRR and a 6.5% increase in Hit@1), with similar results observed on the WN18RR dataset.
2. Scalability: The VAE framework enables the model to perform inference on large-scale knowledge graphs, demonstrating good scalability.

**Weaknesses:**

Weaknesses
1. Model motivation: SBM is well-known graph clustering algorithm. There are many works that combine GNN with SBM to achieve community detection. Except from the used data structure, the key difference between the proposed method and existing works is not obvious.
2. Model Complexity: The introduction of various latent feature sampling and inference mechanisms makes the inference process relatively complex, necessitating more efficient training strategies to speed up training and reduce memory usage.
3. Limited Effectiveness: As noted in the paper, the proposed approach is primarily effective for sparse latent features, with limited performance in dense scenarios. Furthermore, some clustering-based KGE methods should be included in compared baselines.

**Questions:**

1. How to determine the number of clusters in a KG?
2. How to find the specific meaning represented by each cluster?

---

> ### Author Response · Authors · 2024-11-17
> **Response to Weakness 1 & 2**
>
> We are grateful to the reviewer for their insightful feedback and helpful suggestions concerning motivation, model complexity, and effectiveness. In the following, we respond to each comment individually and highlight the corresponding changes made to the revised manuscript.
>
> ## Weakness 1
> > Model motivation: SBM is well-known graph clustering algorithm. There are many works that combine GNN with SBM to achieve community detection. Except from the used data structure, the key difference between the proposed method and existing works is not obvious.
>
> ### Response
> We understand the reviewer’s concerns regarding the **motivation and contributions** of our work. In the following content, we will make clarification on these two points, the corresponding revisions of which in the manuscript will also be mentioned.
>
> **Motivation**
> 1. **Connections among node communities**—a key inductive feature of graph data—have not yet been leveraged for KGC. While a few existing methods incorporate node clustering information, they typically rely on **separate clustering modules** like KMeans, resulting in **suboptimal KGC performance, limited cluster interactions, and a lack of end-to-end design**. To clarify this motivation, we have bolded the relevant content in the Introduction (Line 40, Page 1) and added a dedicated section in Related Work on KGC methods that use neighborhood information (Line 478, Page 9).
>
> 2. Although SBM and VAE have proven effective in statistical link prediction (LP) for detecting node communities, there are two significant differences between traditional LP and KGC: (1) LP methods predict the presence of a link between two nodes, while **KGC requires multi-label edge prediction** and evaluates models based on **tail entity prediction** given a head-relation pair; and (2) modern **KGs are far larger, containing millions of nodes**, unlike typical LP datasets like Cora with only thousands. As such, **directly applying SBM and VAE to KGC is challenging** (Line 82, Page 2).
>
> **Contributions**
>
> 1. We propose **a novel probabilistic generative model for KGC** based on SBM, which enables the integration of complex and meaningful latent structures within KGs in future research.
>
> 2. Our implementation **combines techniques from VAE and text-based KGC methods**, ensuring both **high performance and scalability**. Additionally, the inclusion of text information introduces a fresh approach to clustering within KGs.
>
> 3. We conduct extensive experiments demonstrating our method’s **robust performance, interpretability, and highlighting certain limitations** of leveraging clustering information.
>
> We have revised the **end of the Introduction (Line 84, Page 2)** to provide a more direct and clear listing of our contributions.
>
> ## Weakness 2
> > Model Complexity: The introduction of various latent feature sampling and inference mechanisms makes the inference process relatively complex, necessitating more efficient training strategies to speed up training and reduce memory usage.
>
> ### Response
> We appreciate the reviewer’s emphasis on the importance of time complexity in model evaluation. In the revised manuscript, we have made the following enhancements to address this concern:
>
> 1. We provide a detailed **analysis of the time and space complexity** involved in computing the ELBO (refer to Lines 281, Page 6 and Lines 796, Page 15).
>
> 2. We have also added a **new table (Table 7, Line 930, Page 18) comparing the parameter count and training time** of our method against the baseline model, SimKGC. The results indicate that introducing SBM and VAE **only increases the parameter count by 1.4M (0.8%)**. While the ELBO training requires **additional epochs to converge**, resulting in a slight increase in total training time, we find this increase to be **within an managable range**.

---

> > ### Author Response · Authors · 2024-11-17
> > **Response to Weakness 3 and Questions**
> >
> > ## Weakness 3
> > > Limited Effectiveness: As noted in the paper, the proposed approach is primarily effective for sparse latent features, with limited performance in dense scenarios. Furthermore, some clustering-based KGE methods should be included in compared baselines.
> >
> > ### Response
> > 1. **Limited Effectiveness on FB15k-237 dataset**. Regarding the relatively poorer performance on FB15k-237, we have dedicated substantial effort to **analyzing this issue in the original submission**, covering the discussion in detail from Section 5.2 (MAIN RESULTS) to Section 6 (ANALYSIS). In summary, the dense connections and highly correlated relations in FB15k-237 lead to **fewer distinguishable clustering patterns**, making it challenging for our model to extract cluster-based features. Despite this, our method demonstrates **significant improvements on WN18RR and Wikidata5M and achieves a 2-3 point gain over text-based methods on FB15k-237**.
> >
> > This effectiveness issue **does not undermine the contributions** of our work:
> > - Substantial improvement in KGC performance on general KGs, such as WN18RR and Wikidata5M.
> > - Comprehensive analysis of the difficulties in leveraging clustering information for KGC on highly dense KGs like FB15k-237.
> >
> > 2. **Lack of clustering-based KGE baselines**. We conducted an extensive review of clustering-based KGC methods; however, we found that **relevant literature in this area is limited**. Among the closest works are CTranR [1] and EL-Trans [2], but **neither of these studies reports results on the datasets** used in our paper, which are among the most commonly utilized benchmarks in the field. Furthermore, **no available source code** is publicly available to reproduce their results. As a result, we were unable to include these methods in our experimental comparisons of KGC performance.
> >
> > To clarify the difference between our work and clustering-based methods, we have **added a dedicated section in Related Work** focusing on this line of methods (Line 478, Page 9). focusing on these approaches. Furthermore, we have expanded this section to **include GNN-based methods that utilize neighborhood information in KGs**, following the [feedback of the reviewer 2RBB](https://openreview.net/forum?id=KkVV561IMb&noteId=sjezTkzzQ0).
> >
> > ## Question 1 & 2
> > > How to determine the number of clusters in a KG? How to find the specific meaning represented by each cluster?
> >
> > ### Response
> > 1. **Determining the Number of Clusters**. We adopt a nonparametric Bayesian approach where **the number of clusters is data-driven**. In our work, a large truncation level K (fixed at 128 in our experiments) is preset, providing sufficient clusters for the model. After training, unused clusters (those without nodes) are pruned, and the remaining ones represent the effective number of clusters.
> >
> > 2. **Interpreting Cluster Meaning**. Like most clustering methods, **understanding a cluster’s meaning requires analyzing patterns within the cluster**. To enhance this process, **we incorporate textual features** from the KG into the clustering, making it easier to infer cluster semantics. Additionally, **LLMs can be used** to summarize or interpret the shared attributes of nodes within each cluster, aiding semantic understanding and reducing human intervention.
> >
> > [1] Learning entity and relation embeddings for knowledge graph completion.
> >
> > [2] Knowledge graph embedding and completion based on entity community and local importance.

---

> > > ### Comment · Reviewer_8T3f · 2024-11-17
> > > **Comment**
> > >
> > > Thanks for your rebuttal. I have raised my score to 5, but I still think the model motivation is not impressive.

---

### Official Review · Reviewer_yn9h · 2024-10-31

**Soundness:** 3
**Presentation:** 3
**Contribution:** 3
**Rating:** 6
**Confidence:** 3

**Summary:**

This paper introduces DSLFM-KGC, a novel framework of sparse latent feature models for KGC. The framework leverages stochastic blockmodels (SBMs) and deep variational autoencoders (VAEs) to capture latent community structures in KGs and improve link prediction. Experimental results show the effectiveness of DSLFM-KGC.

**Strengths:**

1.The paper is clearly written and easy to follow.

2. The theoretical foundation of this paper is solid, utilizing extensive mathematical formulations to elucidate the structure of the proposed model or to demonstrate its validity.

3. The experiments conducted on benchmark datasets demonstrate the effectiveness of DSLFM-KGC in improving KGC performance and uncovering interpretable latent structures.

**Weaknesses:**

1.The paper repeatedly emphasizes the advantages of the proposed model on large-scale graphs; therefore, it would be beneficial to include comparative experiments on time complexity or runtime performance between the proposed model and the baseline.

2. At the end of the introduction section, a more direct listing of the contributions of this paper should be provided, with particular emphasis on the novel points introduced for the first time in this work.

3. The paper uses more baselines on WN18RR and FB15k-237 compared to Wikidata5M. Is this due to differences in dataset scale? The authors should provide a detailed explanation.

4. The primary parts in this framework, SBM and VAE, are derived from previous work, with extensive references to existing literature in the framework description. I am not entirely clear on the main innovations introduced by the authors in these two methods. It would be helpful if the authors could enhance the explanation of their contributions in the text or directly clarify these innovations in their response to me.

**Questions:**

Please refer to the "weaknesses" section.

---

> ### Author Response · Authors · 2024-11-15
> **Response Addressing Model Complexity, Contributions, and Baseline Issues**
>
> We would like to thank the reviewer for the thoughtful feedback and helpful suggestions on representation, clarity and experiment settings. Below, we provide a point-by-point response to the reviewer’ comments, with references to the changes made in the manuscript.
>
> ## Weakness 1
> > The paper repeatedly emphasizes the advantages of the proposed model on large-scale graphs; therefore, it would be beneficial to include comparative experiments on time complexity or runtime performance between the proposed model and the baseline.
>
> ### Response
> We appreciate the reviewer’s emphasis on the importance of time complexity in model evaluation. In the revised manuscript, we have made the following enhancements to address this concern:
>
> 1. We provide a detailed **analysis of the time and space complexity** involved in computing the ELBO (refer to Lines 281, Page 6 and Lines 796, Page 15).
>
> 2. We have also added a **new table (Table 7, Line 930, Page 18) comparing the parameter count and training time** of our method against the baseline model, SimKGC. The results indicate that introducing SBM and VAE **only increases the parameter count by 1.4M (0.8%)**. While the ELBO training requires **additional epochs to converge**, resulting in a slight increase in total training time, we find this increase to be **within an managable range**.
>
> ## Weakness 2
> > At the end of the introduction section, a more direct listing of the contributions of this paper should be provided, with particular emphasis on the novel points introduced for the first time in this work.
>
> ### Response
> Thank you for your valuable feedback. We have revised the end of the Introduction (Line 84, Page 2) to provide a more direct and clear listing of our contributions.
>
> ## Weakness 3
> > The paper uses more baselines on WN18RR and FB15k-237 compared to Wikidata5M. Is this due to differences in dataset scale? The authors should provide a detailed explanation.
>
> ### Response
> We understand the reviewer’s concerns regarding the fairness and consistency of using different numbers of baselines across datasets. **The Wikidata5M dataset is primarily used by text-based methods**, and KGC performance results on this dataset are **rarely reported in embedding-based literature**. Therefore, to ensure meaningful comparisons, we selected only text-based baselines on the Wikidata5M dataset to evaluate our method’s performance.
>
> ## Weakness 4
> > The primary parts in this framework, SBM and VAE, are derived from previous work, with extensive references to existing literature in the framework description. I am not entirely clear on the main innovations introduced by the authors in these two methods. It would be helpful if the authors could enhance the explanation of their contributions in the text or directly clarify these innovations in their response to me.
>
> ### Response
> We understand the reviewer’s concerns regarding the **motivation and contributions** of our work. In the following content, we will make clarification on these two points, the corresponding revisions of which in the manuscript will also be mentioned.
>
> **Motivation**
> 1. **Connections among node communities**—a key inductive feature of graph data—have not yet been leveraged for KGC. While a few existing methods incorporate node clustering information, they typically rely on **separate clustering modules** like KMeans, resulting in **suboptimal KGC performance, limited cluster interactions, and a lack of end-to-end design**. To clarify this motivation, we have bolded the relevant content in the Introduction (Line 40, Page 1) and added a dedicated section in Related Work on KGC methods that use neighborhood information (Line 478, Page 9).
>
> 2. Although SBM and VAE have proven effective in statistical link prediction (LP) for detecting node communities, there are two significant differences between traditional LP and KGC: (1) LP methods predict the presence of a link between two nodes, while **KGC requires multi-label edge prediction** and evaluates models based on **tail entity prediction** given a head-relation pair; and (2) modern **KGs are far larger, containing millions of nodes**, unlike typical LP datasets like Cora with only thousands. As such, **directly applying SBM and VAE to KGC is challenging** (Line 82, Page 2).
>
> **Contributions**
>
> 1. We propose **a novel probabilistic generative model for KGC** based on SBM, which enables the integration of complex and meaningful latent structures within KGs in future research.
>
> 2. Our implementation **combines techniques from VAE and text-based KGC methods**, ensuring both **high performance and scalability**. Additionally, the inclusion of text information introduces a fresh approach to clustering within KGs.
>
> 3. We conduct extensive experiments demonstrating our method’s **robust performance, interpretability, and highlighting certain challenges** of leveraging clustering information.

---

> > ### Author Response · Authors · 2024-11-22
> > **Gentle Reminder: Final Discussion Stage**
> >
> > Dear Reviewer yn9h,
> >
> > We sincerely appreciate your thoughtful comments, which have significantly contributed to improving our paper. As the discussion phase nears its conclusion, we wanted to send a gentle reminder to ask if you might have any additional questions or points of clarification regarding our submission. We are happy to address them while there is still time.

---

> > > ### Comment · Reviewer_yn9h · 2024-11-22
> > >
> > > Thanks for your response. I would like to hold my score.

---

### Official Review · Reviewer_XAos · 2024-11-04

**Soundness:** 2
**Presentation:** 1
**Contribution:** 2
**Rating:** 3
**Confidence:** 5

**Summary:**

This paper addresses the limitations of recent text-based knowledge graph completion (KGC) methods that fail to adequately consider the complex interconnections among entities in large-scale knowledge graphs. It introduces a novel framework utilizing sparse latent feature models optimized via a deep variational autoencoder (VAE), which enhances link prediction and offers interpretability of latent structures by integrating textual information. Experimental results on multiple datasets demonstrate that the proposed method significantly improves performance by uncovering latent communities and generating interpretable representations.

**Strengths:**

1. The proposed method aims to balance the retention of critical knowledge with the elimination of redundancy, which is an interesting topic.
2. The authors not only effectively complete missing triples but also provide clear interpretability of the latent structures which seems reasonable.

**Weaknesses:**

1. The paper is not organized clearly, which is not friendly for understanding. For example, there is a lack of preliminary details for how to model MB and other module in 3.1 GENERATIVE MODEL.
2. Figure 2 lacks of explanation, \textit{e.g.,} how the modules work together and match the equations in the main paper. The paper lacks the necessary reproduction file for the results.
3. The paper lacks the analysis of time complexity as well as space complexity, which is necessary to study the efficiency of the model.
4. The authors do not compare the model with other SOTA KGE methods, e.g.,[1][2][3]. The performance of, MRR in FB15K-237 is 0.36 while that of the proposed paper is 0.355. In this way, the performance of the proposed paper is not significant and the authors may better give a reasonable explanation.
[1] Compounding Geometric Operations for Knowledge Graph Completion
[2] Geometry interaction knowledge graph embeddings
[3] KRACL: Contrastive Learning with Graph Context Modeling for Sparse Knowledge Graph Completion

**Questions:**

Please refer to Weaknesses.

---

> ### Author Response · Authors · 2024-11-15
>
> We sincerely thank the reviewer for the thoughtful and constructive feedback, which has helped us improve the clarity, rigor, and quality of our manuscript. Below, we provide point-by-point responses to the comments and outline the corresponding changes made to the paper.
>
> ## Weaknesses 1
> > The paper is not organized clearly, which is not friendly for understanding. For example, there is a lack of preliminary details for how to model MB and other module in 3.1 GENERATIVE MODEL.
>
> ### Response
> We thank the reviewer for highlighting the issues regarding organization and clarity.
>
> First of all, we would like to clarify the modeling described in Section 3.1 GENERATIVE MODEL, where we present a **probabilistic framework**. The “modules” described in this section correspond to probabilistic distributions assumed for the variables. For instance, Equation 4,  $w_{hr} \sim \mathcal{N}(w_{hr} | \mathbf{0}, \sigma^2 \mathbf{I})$ means that $w_{hr}$ follows a Gaussian distribution. Similarly, the "$\mathcal{MB}$ module" refers to a multivariate Bernoulli distribution, as defined in the Preliminary section (Equation 1)
>
> We have structured the modeling and implementation details as follows:
> -  **In Section 3.1**, we describe the **probabilistic distributions for latent variables**, such as $p(z_{hr}) and p(w_{hr})$, as well as for observed triples $A_{hr, t}$ in Equation 7.
> - **In Section 3.2**, we explain how to **learn the posterior distributions** of the latent variables $\mathcal{H}$ (Equation 11 to 15) using a VAE encoder (Equations 11–15)
> - **In Section 3.3**, we describe the decoding process, which **uses the latent variables to complete missing triples**.
>
> Thus, Section 3.1 introduces the core probabilistic modeling, while Sections 3.2 and 3.3 detail the implementation steps for encoding and decoding. We hope this clarifies the structure and flow of the manuscript.
>
> ## Weaknesses 2
> > Figure 2 lacks of explanation, \textit{e.g.,} how the modules work together and match the equations in the main paper. The paper lacks the necessary reproduction file for the results.
>
> ### Response
> We agree that Figure 2 would benefit from more detailed explanations. We have **updated the caption of Figure 2** to include a comprehensive description of how the modules work together.
>
> We have prepared a detailed supplementary material file, including all the necessary reproduction files, such as datasets, code, and instructions for running the experiments.
>
> ## Weaknesses 3
> > The paper lacks the analysis of time complexity as well as space complexity, which is necessary to study the efficiency of the model.
>
> ### Response
> We appreciate the reviewer’s emphasis on the importance of time andspace complexity in model evaluation. In the revised manuscript, we have made the following enhancements to address this concern:
>
> 1. We provide a detailed **analysis of the time and space complexity** involved in computing the ELBO (Lines 281, Page 6 and Lines 796, Page 15).
>
> 2. We have also added a **new table (Table 7, Line 930, Page 18) comparing the parameter count and training time** of our method against the baseline model, SimKGC. The results indicate that introducing SBM and VAE **only increases the parameter count by 1.4M (0.8%)**. While the ELBO training requires **additional epochs to converge**, resulting in a slight increase in total training time, we find this increase to be **within an managable range**.
>
> ## Weaknesses 4
> > The authors do not compare the model with other SOTA KGE methods, e.g.,[1][2][3]. The performance of, MRR in FB15K-237 is 0.36 while that of the proposed paper is 0.355. In this way, the performance of the proposed paper is not significant and the authors may better give a reasonable explanation.
>
> ### Response
> Thank you for pointing out the need for comparison with SOTA methods.
>
> 1. **Baseline Inclusion**. In the revised manuscript, we **have included the SOTA results from KRACL [3]**  for its best performance on WN18RR, as shown in Table 1. However, we would like to clarify that the **original submission already included several SOTA KGE methods, such as HittER and N-Former**, which achieve strong performance on FB15k-237, with MRR scores of 37.3 and 37.2, respectively. The inclusion of the additional baseline **does not alter the overall comparative assessment presented in our paper**.
>
> 2. **Performance on FB15k-237**. Regarding the relatively poorer performance on FB15k-237, we have dedicated substantial effort to **analyzing this issue in the original submission**, covering the discussion in detail **from Section 5.2 (MAIN RESULTS) to Section 6 (ANALYSIS)**. In summary, the dense connections and highly correlated relations in FB15k-237 lead to **fewer distinguishable clustering patterns**, making it challenging for our model to extract cluster-based features. Despite this, our method demonstrates **significant improvements on WN18RR and Wikidata5M and achieves a 2-3 point gain over text-based methods on FB15k-237**.

---

> > ### Author Response · Authors · 2024-11-22
> > **Gentle Reminder: Final Discussion Stage**
> >
> > Dear Reviewer XAos,
> >
> > We sincerely appreciate your thoughtful comments, which have significantly contributed to improving our paper. As the discussion phase nears its conclusion, we wanted to send a gentle reminder to ask if you might have any additional questions or points of clarification regarding our submission. We are happy to address them while there is still time.

---

### Meta-Review · Area_Chair_jrYi · 2024-12-17

**Metareview:**

Given that Stochastic blockmodels (SBMs) offer probabilistic frameworks for capturing latent community structures, this manuscript proposes a sparse latent feature model for knowledge graph completion(KGC) optimized through a deep variational autoencoder(VAE). Experiments on the WN18RR, FB15k-237, and Wikidata5M datasets show that the proposed method reveals latent communities and produces interpretable representations.

Strengths: the idea of relating the community structure and improving link prediction performance is reasonable, and the experimental results support the effectiveness of the proposed approach.

Weaknesses: the manuscript would benefit from more explicit descriptions of the motivation and distinctions between the proposed method and the existing approaches, as pointed out by Reviewers 2RBB and 8T3f. Also, more direct intuition examples should be provided to clarify why clustering information is critical for KGC. Moreover, the manuscript would benefit from comparisons to typical GNN-based and LLM-based models, e.g., the pros and cons of the proposed method compared to these methods. The latter is needed because the authors claim that interpretable representations are one of the proposed method's benefits.

Overall, the manuscript needs major revisions before it can be ready for publication.

**Additional Comments On Reviewer Discussion:**

Reviewer 2RBB and Reviewer 8T3f have provided helpful feedback for me to make the final decision. I agree that the manuscript should describe the motivation more clearly and provide supporting examples showing the direct relationship between uncovering the community structure and improving the link prediction performance. Also, the manuscript should explicitly describe the distinctions between the proposed approach and the existing works. Furthermore, some comparisons to GNN-based and LLM-based models are needed to fully support the benefits of the proposed method.

---

### Decision · Program_Chairs · 2025-01-22

Reject